# Development and Quality Evaluation of Rigatoni Pasta Enriched with Hemp Seed Meal

**DOI:** 10.3390/foods12091774

**Published:** 2023-04-25

**Authors:** Marina Axentii, Silviu-Gabriel Stroe, Georgiana Gabriela Codină

**Affiliations:** Faculty of Food Engineering, Stefan cel Mare University of Suceava, 720229 Suceava, Romania; axentii.marina@outlook.com (M.A.); silvius@fia.usv.ro (S.-G.S.)

**Keywords:** hemp seed meal (HSM), pasta, product development, quality characteristics

## Abstract

Existing food trends and modern consumers’ nutritional preferences have led to a rising demand for plant-based sources of protein such as hemp seed meal and the possibility of consumption hemp-rich products, most often in wheat-based staple foods, such as pasta. Pasta, as a conventional food product, is widely consumed worldwide due to its nutritional value, long shelf life, easy preparation, versatility of uses and also relatively low cost, which has made this product popular over time. Five formulations of rigatoni-shaped pasta obtained by partial replacement of wheat grain flour with 5%, 10%, 15% and 20% hemp seed meal (HSM) were studied regarding the technological, physicochemical, textural, antioxidant and sensory properties of the pasta samples. The substitution of wheat flour with hemp seed meal (HSM) led to a slight increase in the cooking loss (CL) and optimal cooking time (OCT) compared to the control sample, while the water absorption (WA) and swelling index (SI) decreased during evaluation. The experimental results also showed a decrease in luminosity and fracturability, with an increased firmness of pasta dough. Moreover, the developed pasta showed a significant improvement in antioxidant capacity in terms of total phenolic content (TPC) and antioxidant activity (DPPH). The pasta samples with 15% and 20% HSM substitutes experienced a browning process due to the Maillard reactions during drying, as well as a color loss during cooking; however, the color changes did not affect the acceptability of the product. The partial replacement of wheat flour with hemp seed protein highlighted the possibility of developing a new innovative type of pasta that claims a functional benefit and presents an improved nutritional value, mainly due to the partial protein intake, as well as certain benefits for a human diet.

## 1. Introduction

Globally, pasta consumption and sales reached USD 22.8 Billion in 2022, and this value is expected to grow fluently as consumers are more conscious of the environment and its major climate changes, but also the trendy adoption of a vegan or vegetarian health-promoting diet significantly decreased the demand for meat-based food products [1]. Pasta represents one of the most accessible and carbohydrate-rich food products all over the world [2]. From a manufacturer’s perspective, pasta can be considered a versatile and economically convenient food product, easy to produce, transport, stock and consume. It has a long shelf life, and its recipe can be easily reformulated or improved considering the consumer’s preferences and their nutritional needs. Due to the continuous rise of the pasta market size, food industries are interested in doing as much research as possible based on how healthy pasta can be by improving it with different inclusions of nonconventional and sometimes atypical ingredients or by-products considering the rising demand of health-conscious consumers for functional foods [3]. As traditional wheat pasta is relatively poor in proteins and other physiologically active compounds, many studies focus on looking for different methods of increasing pasta’s nutritional value [4]. There are several important studies on innovative pasta formulations produced by partial replacement of wheat grain flour with various plant-based sources of protein, such as chickpeas [5], hemp [6], soy [7], lentils [8], spirulina [9], mushroom powder [10], quinoa protein isolate [11] and different species of algae, that present a primary interest for pasta manufacturers, who aims to satisfy the high nutritional standards of modern consumers by raising the nutritional value of the product [12]. For such purpose, the fortification of cereal food products, such as pasta, with pseudo-cereal and legume ingredients promises a substantial development of well-balanced protein diets, as they represent a rich source of amino acids that complements the nutritional value of the functional products [13].

For the functional foods segment—which includes such products as nutrition bars, meal replacers and supplements, and innovative food products in general—protein content and its quality is one of the most important requirements [14].

As the nutritional needs of the consumers are mostly focused on the quality of those amino acids that a plant-based protein can provide, hemp seed protein can be considered one of the most biologically suitable protein substitutes for a meat-type protein [15]. Chemically, the protein fraction represents almost a quarter of the total hemp composition. Hemp seed protein consists mainly of globulins, in proportions of 65–75%, albumins (25–37%) and not least proteins rich in sulfur. However, the most valuable protein that makes this ingredient an important plant-based source of protein is edestin, which is chemically a globulin type protein rich in arginine with high digestibility that is also considered to be the “backbone of human DNA” [16]. Hemp protein contains the same amount of protein as mutton or beef products. In comparison with other vegetable sources of proteins, hemp seed meal has a balanced amino acid profile, high fiber content and bioactive peptides with strong antioxidant behavior. The consumption of hemp-rich products is therefore recommended for people that suffer from celiac disease, as hemp seeds do not contain gluten, athletes who desire a rich protein diet, children, because it meets the recommended dietary allowance (RDA) of children protein intake, according to the Food and Agriculture Organization (FAO), lactose intolerant people and other consumers who require a healthy lifestyle on a daily basis. Moreover, hemp seed protein contains no known allergens, making it hypoallergenic, and therefore 100% safe for human food use. Being cultivated for thousands of years, hemp is sustainably grown without any use of chemicals such as herbicides, pesticides or fungicides, which all greatly impact the environment [17]. Furthermore, as hemp contains all nine amino acids, being considered a complete plant-based protein, it is perfectly suitable for a vegan or vegetarian diet [18].

Therefore, hemp-rich products, especially pasta enriched with hemp seed meal (HSM) contribute to a healthy, complete and nutritionally balanced diet, helping to maintain your body and keep your mind active.

The importance of this research is reflected in the consumer’s demand oriented towards functional products and protein-enriched products. Moreover, the work supports food processors with useful information related to the technological behavior of the dough and hemp pasta samples while cooking, as the final product quality depends on its behavior during processing. Although the nutritional needs of the consumers can be satisfied by formulating a new type of functional pasta, the feasibility of the product itself might suffer some significant difficulties, especially in terms of technological properties and several noticeable consequences of heat processing [19].

The present research aims to study the effect of the partial replacement of wheat grain flour with hemp seed meal (HSM) on a new assortment of pasta, which has a functional benefit, regarding the technological, rheological, textural and sensory properties of pasta samples.

## 2. Materials and Methods

### 2.1. Materials

The main raw materials used for rigatoni pasta were wheat flour of 650 type (SC Mopan Suceava, Romania) and hemp seed protein powder (Canah, Romania) which was produced by grinding down to a fine powder the hemp seed “meal” also called hemp “cake”, a by-product resulted from hemp oil extraction [2]. According to the manufacturer, HSM has a medium particle size, which is approximately 200–236 µm. The wheat flour, hemp seed meal (HSM) and the pasta chemical composition were analyzed according to the ICC standard methods: ash content (ICC 104/1), protein content (ICC 105/2), fat content (ICC 136) and moisture content (ICC 110/1). For the wheat flour, the wet gluten content (ISO 21415-2:2015) was also determined, as was the falling number value (ICC 107/1 method) and the gluten deformation index (SR 90/2007) according to a Romanian standard method.

### 2.2. Pasta Formulation

Wheat flour was substituted with different levels of hemp seed meal (HSM) as following: 0%—control sample, 5%—HSM_5, 10%—HSM_10, 15%—HSM_15, 20%—HSM_20. The dough of the pasta samples was formulated by mixing all ingredients, using a Kitchen Aid (Whirlpool Corporation, Benton Harbor, MI, USA) mixer. The hydration levels used were 58.7% for the control sample, 59.9% for HSM_5, with 5% wheat substitution, 60.4% for HSM_10 (10%), 61% for HSM_15 (15%) and 61,4% for HSM_20 with 20% of wheat substituted with hemp seed meal (HSM), respectively. Each sample was also made using 2 g of salt The process itself included several steps: first, the mixing of all solid ingredients, such as wheat flour, hemp seed meal and salt, followed by adding water and mixing again for 3–5 min using a dough hook until all ingredients were evenly combined. The resulted firm dough ball was extruded using a KitchenAid pasta press with a rigatoni mold. Rigatoni-shaped pasta was dried in a functional dryer with a heat ventilator unit to ensure uniform temperature and ventilation in all parts of the dryer and a moisture control unit for the constant humidity at 40 °C for 16 h. After a complete drying circle, pasta was cooled for 2 h at room temperature and subsequently stored in plastic bags with zip locks until further experiments were performed.

### 2.3. Dough Rheological Analysis

In order to obtain the rheological extension data of pasta dough, an Alveograph device (Chopin Technologies, Cedex, France) was used following the standard methods procedures AACC 54-30A, ICC 121 or ISO 5530/4. In essence, a mixture of wheat flour, hemp seed meal and salty water mixed into a dough ball is supposed to resist a biaxial extension, which is generated by a pneumatic circuit, forming a rising bubble of dough due to the gluten network, which is continuously filling with air until it breaks at a critical point/value. The following parameters were measured: dough extensibility (L), baking strength (W), maximum pressure (P) and the configuration ratio of the Alveograph curve (P/L ratio) of the pasta dough [20].

#### Dynamic Dough Rheological Evaluation

The dynamic dough rheological properties were evaluated using a Thermo-HAAKE, MARS 40 rheometer (Karlsruhe, Germany). The frequency sweep test was performed on laminated dough samples, rested in advance for 30 min at room temperature for internal strain elimination. Storage modulus (G’) (elastic property), loss modulus (G”) (viscous property) and loss tangent (tan δ) were evaluated. Each pasta sample prepared to the optimum dough hydration level was analyzed using two parallel plate geometry (d = 40 mm) and a gap of 2 mm. During the sweep test, frequency varied from 0.1 to 20 Hz, at a constant strain of 15 Pa. For the temperature sweep test, the dough samples were heated from 25 °C to 100 °C at a rate of 4 °C per min. For this purpose, the initial gelatinization temperature (T_i_) and maximum gelatinization temperature (T_max_) were analyzed.

### 2.4. Dough Microstructure Analysis

The morphology of the dry pasta surfaces was determined using a scanning electron microscope (SEM) Hitachi SU-70 (Tokyo, Japan) equipped with a detector of secondary electrons (SE) of the Everhart-Thornley type, which requires a high vacuum environment. The samples were metallized with a layer of 5 nm of Ag. The acceleration voltage was 5.00 kV, and the images were obtained at a 1000× magnification.

### 2.5. Pasta Technological Characteristics

The technological parameters of the rigatoni pasta samples that give a better appreciation of pasta cooking quality were determined by the AACC International Approved Method 66-50.01 [2]. All pasta samples were cooked to their optimum cooking time (OCT), to assess texture, cooking loss (CL), water absorption (WA) and swelling index (SI), as described in the standard method.

#### 2.5.1. Optimal Cooking Time

The optimal cooking time (OCT), expressed in minutes, was determined according to the AACC International Approved Method 66-50.01 [2]. For OCT determination, only distilled water was used. The experiment itself involved boiling 10 g of pasta in 200 mL of distilled water, as this amount is enough to cover the pasta completely in a 500 mL vessel, and timing each 30 to 30 s until the experiment was complete. It was experimentally considered that the pasta had reached the optimal cooking time when all white particles of starch in the cooked pasta disappeared after compressing it between two glass plates at each 30 s [12].

#### 2.5.2. Water Absorption (WA)

An amount of 10 g of pasta sample were boiled in 200 mL of distilled water for a certain period of time equivalent to the optimal cooking time (OCT) of each pasta sample, followed by draining the excess water using a fine sieve for 3 min and weighting the resulted cooked product. The water absorption index was determined with Equation (1):WA = [(w_1_ − w_2_)/w_2_] × 100(1)
where w_1_ is the weight of the cooked pasta and w_2_ is the weight of the pasta after drying [21].

#### 2.5.3. Cooking Loss (CL)

Cooking loss was determined using the water collected from the previous determination of water absorption index (WAI) by drying the whole amount of collected water in a heat-resistant glass vessel in an oven at 105 °C until total liquid evaporation and weighting the solid residue. Cooking loss (CL) represents, therefore, the amount of solids lost into the cooking water, expressed in grams/100 g of raw pasta [22].

#### 2.5.4. Swelling Index (SI)

Swelling Index was obtained by drying cooked pasta, previously boiled according to its OCT, until a constant weight at 105 °C (approximately 16 h), then cooled at room temperature and weighed again. The final results were calculated with Equation (2):SI = (m_1_ − m_2_)/m_2_(2)
where m_1_ is the weight of the cooked pasta and m_2_ is the weight of the pasta after drying [23].

### 2.6. Color Pasta Characteristics

Dry and cooked pasta color parameters were measured using a Minolta Chromameter (Model CR-400, Minolta Co., Osaka, Japan). In order to obtain precise and consistent color evaluation CIEL*a*b* color space was used, where in terms of colorimetric coordinates, L* represents the lightness while a* and b* represent the green to red and blue to yellow color ranges, respectively [24].

### 2.7. Textural Pasta Characteristics

Pasta textural parameters were performed using a texture analyzer TVT-6700 (Perten Instruments, Hägersten, Sweden) equipped with a 10 kg load cell, as in the method described by Iuga, 2020 [25]. An amount of 50 g of freshly made dough of each pasta sample was submitted to double compression using a 25-mm diameter cylindrical probe, a trigger force of 20 g, at a speed of 1 mm/s up to 50% of the original height. The following textural parameters were determined: firmness, cohesiveness, adhesiveness and elasticity [26].

### 2.8. Dry Pasta Fracturability

Dry pasta was submitted to a fracturability test that measured the maximum force F (g) required to break a dry pasta sample into multiple pieces using a Perten TVT-6700 device (Perten Instruments, Hägersten, Sweden). For this purpose, an aluminum break rig set was use, adjusted to 13 mm width, with a trigger force of 50 g at a speed set to 3 mm/s [9]. For the fracturability test, dry pasta samples with a 16 mm diameter that were 45 mm long were used.

### 2.9. Pasta Total Phenolic Content and Antioxidant Activity

Extraction and determination of total phenolic content (TPC) were performed according to the methods described in [22,27], on pasta products, with some modifications. All pasta samples were first milled to a thin powder using a grinder (Myria 4053RD) in order to obtain a better extraction of polyphenolic compounds. Extraction itself was carried out by mixing 2 g of each pasta sample with 20 mL of methanol 80% (*v/v*) in a sonication bath for 40 min at 37 °C and 45 kHz, followed by a double filtration. The determination of total phenolic content (TPC), expressed in μg GAE/g, was made using the Folin–Ciocalteu method as follows: 0.2 mL of extract (diluted 1:5 with distilled water) was mixed with 2 mL of Folin–Ciocalteu reagent (1N) (diluted 1:10 with distilled water) and 1.8 mL of sodium carbonate 20% (*w/v*) into a tube. The solution was left in the dark at room temperature for 30 min. The TPC determination was performed at 750 nm wavelength on a UV–VIS–NIR Shimadzu 3600 (Tokyo, Japan) spectrophotometer. Each sample was evaluated in triplicate, and TPC values were calculated through extrapolation using a calibration curve (R2 = 0.99) made with different concentrations of gallic acid (GAE).

Pasta antioxidant activity was assessed by reading the absorbance of the solutions at 517 nm wavelength using a UV-VIS-NIR spectrophotometer, as in the method described by Chetrariu [27], as follows: 2 mL of the extract described above was mixed with 2 mL of 2,2-Diphenyl-1-picrylhydrazy solution (DPPH) 0.1 mM into a tube. The solution was agitated for 2 min and left in the dark at room temperature for 30 min. The measurements for each sample were performed in triplicate, using distilled water as a blank sample, according to the following equation:%Inhibition of DPPH = [(1 − A_s_/A_b_)] × 100(3)
where A_s_ is the absorbance of the sample and A_b_ is the absorbance of the blank sample, both read at the same wavelength (λ = 517 nm).

### 2.10. Pasta Sensory Characteristics

The sensory profile of the pasta samples was evaluated using 30 semi-trained panelists. The sensory evaluation was performed at the Sensory Laboratory Lab of the Faculty of Food Engineering, Stefan cel Mare University, Suceava, Romania. Each panelist evaluated the sensory characteristics of the pasta samples, which were placed on white plates codified with black three-digit numbers, which did not contain the letter A or the number 1. Before tasting, each panelist cleaned their palate with water. For sensory analysis, the hedonic test was used with a nine-point hedonic scale (1—“dislike extremely” to 9—“like extremely”). The sensory pasta characteristics evaluated were texture, taste, smell, color, aroma, aspect and general acceptability.

### 2.11. Statistical Analysis

All experimental data obtained were processed using the Statistical Package for Social Science (v.16, SPSS, Chicago, IL, USA). The experimental data were expressed as the mean value ± standard deviation. The statistical differences between means were analyzed by applying ANOVA analysis of variance, with Tukey’s test at a 5% significance level (*p* < 0.05) [15]. The relations between pasta quality and technological characteristics may be seen through Principal Component Analysis (PCA) obtained using XLSTAT for Excel 2023 free trial version software.

## 3. Results

### 3.1. Flour Characteristics

The wheat flour that was used for pasta formulations presented the following data: 0.65% ash content, 1.17% fat, 30.2% wet gluten, 11.9% protein, 14.2% moisture, 4 mm gluten deformation index and 386 s the falling number value. According to the wheat flour data value characteristics, this was of a strong quality for bread making and low α-amylase activity [28]. The hemp seed meal (HSM) presented the following data values: 8.8% moisture, 11.0% fat, 51% protein and 3.9% ash content.

### 3.2. Dough Rheological Characteristics

#### 3.2.1. Empirical Dough Rheological Characteristics

The empirical dough rheological characteristics during the extension of pasta samples are shown in Table 1. The experimental results showed that dough samples with 5% and 10% HSM addition levels presented no significant differences (*p* < 0.05) between Alveograph values maximum pressure, extensibility and P/L ratio. All the dough samples presented high P/L values (˃1), indicating a strong gluten suitable for pasta production [25]. From all the analyzed samples, the sample with 15% HSM presented the highest P/L value, a fact which is expected because it presents the highest tenacity and the lowest extensibility from all the analyzed ones. The baking strength presented the highest values for the samples with 5–15% HSM enrichment in wheat flour, which indicates the fact that it may produce pasta of superior quality. The Alveograph W value is considered a valid one that may predict the pasta quality associated with a strong dough network able to retain starch granules during cooking [29]. Due to the fact that the sample with 15% HSM replacement presents a high baking strength compared to the control sample, a high tenacity and P/L ratio may indicate the fact that, from all the analyzed samples, it presents the optimal dough rheological characteristics necessary to obtain pasta with a striated profile, which must maintain its shape after cooking.

#### 3.2.2. Fundamental Dough Rheological Characteristics

The fundamental rheological characteristics of dough samples formulated by partial replacement of wheat flour with different levels of HSM can be seen in Figure 1 and Figure 2. Figure 1 highlights the frequency sweep data for dough samples with different levels of HSM enrichment, whereas Figure 2 presents the evolution of G’, G” and tan δ of dough samples in which the HSM was incorporated. As can be seen from the frequency sweep data, the HSM enrichment led to a decrease in G’ and G” and an increase in tan δ values, indicating a decrease in dough consistency. For whole ranges of frequencies, the storage modulus (G’) have higher values than the loss ones (G”), indicating a solid elastic-like behavior for all dough samples [30]. The loss tangent is lower than 1 for all the dough samples, indicating the fact that the elastic part of the dough is more prominent than the viscous one. The control sample presented the highest tan δ values compared to the samples with HSM incorporated into the dough recipe. With the increase in HSM substitutions in wheat flour, the tan δ values decreased, meaning that the dough is losing its elasticity, so it is increasing the viscosity.

Figure 2 presents the G’, G” and tan δ variation data as the temperature rises. In this case, the control sample presented the lower values of the G’ and G” moduli and the higher value of tan δ. A similar trend on dough rheological behavior during the temperature increase by hemp meal enrichment has also been reported by Istrate et al., when hemp flour was incorporated into a dough recipe [31]. 

### 3.3. Dough Microstructure

SEM Analysis allows a better understanding of the textural profile of the pasta samples and helps to appreciate the effect of the partial replacement of wheat grain flour with different levels of hemp seed meal on the dough strength, which influences the quality of the final product [32]. Figure 3 presents micrographs of dry pasta samples enriched with different levels of HSM, according to the dough recipe. Several structural changes can be observed: the appearance of cracks on the outer and inner surfaces, the development of alveoli inside the swelling starch granules on the inner surface of the pasta samples, several agglomerations of starch granules, especially in the case of dough samples with 10% and 15% hemp seed meal enrichment and not least irregularly distributed starch granules on the whole area surfaces.

### 3.4. Pasta Quality Evaluation

#### 3.4.1. Pasta Technological Characteristics

The pasta technological characteristics, in terms of optimal cooking time (OCT), water absorption (WA), cooking loss (CL) and swelling index (SI), are shown in Table 2. As can be seen, pasta samples enriched with different levels of HSM showed a significant increase (*p* < 0.05) of the OCT value compared with the control sample but still presented a lower value compared to the manufacturing rigatoni OCT parameters, which were between 11 and 14 min [33]. This is a very important parameter that influences the consistency and firmness of pasta in the course of cooking [11,12,13]. From a sensory point of view, undercooked pasta with a suitable firm texture, referred to as “al dente”, is more appreciated by consumers [10]. A variation of the OCT from 9 to 10 min was registered due to the partial protein replacement of the wheat flour. The same tendency was also reported for pasta protein supplementation with hemp cake and hemp flour [9], as well as with mushroom powder and Bengal gram flour [34]. The partial substitution of wheat flour with hemp seed meal (HSM) in pasta led to an increase in OCT and CL, while WA and SI significantly decreased (*p* < 0.05) compared to the control sample.

#### 3.4.2. Dry and Cooked Pasta Color Characteristics

Pasta color is one of the most important sensory characteristics that considerably influence consumers’ purchasing decisions. At the same time, the color of the food product is unconsciously associated with its aroma and probable taste. The changes in pasta color parameters after the partial substitution of wheat flour with 5%, 10%, 15% and 20% HSM in dry and cooked pasta, which can be seen in Figure 4, are shown in Table 3. Dry pasta presented a significant decrease (*p* < 0.05) in brightness (L*) and an increase in redness (highest a* value), which may be associated with the development of the Maillard reaction [10]. Pasta with 20% wheat substitution with HSM presented the darkest color but also suffered a significant color loss while cooking. Control samples showed lower L* values, mainly due to the partial pigments released in the boiling water after cooking. The partial loss of some compounds in the boiling water is also noticeable by the increased volume of pasta samples followed by a slight discoloration, also suggested by the values of the a* and b* parameters, which are relatively lower than those in the case of the dry pasta.

#### 3.4.3. Pasta Textural Characteristics

Table 4 shows the pasta textural characteristics with different levels of HSM incorporated into the pasta recipe. The evaluation of the textural profile of pasta samples is very important, as the texture primarily influences the general acceptability of the product in terms of sensory characteristics and quality evaluation. No significant changes (*p* < 0.05) were noticed for cohesiveness and stickiness values when high levels of HSM were incorporated into the wheat flour. Compared to the control sample, the firmest pasta sample was the one with 20% HSM incorporated in the dough recipe. In addition, this sample presented the lowest adhesiveness value. Similar data have also been reported by Chetariu and Dabija when spent grains were incorporated into a dough recipe [27].

#### 3.4.4. Dry Pasta Fracturability

The partial replacement of wheat flour with HSM showed a decrease in fracturability compared to the control sample, as can be seen from Table 4. Similar results were obtained by De Marco et al. for dough samples with spirulina addition, probably due to the fact that spirulina, similar to hemp, lowers breaking stress due to a stronger internal structure of the pasta samples [9]. A higher force to break the pasta is desirable [35] as pasta must keep its specific shape while drying and cooking.

#### 3.4.5. Pasta Total Phenolic Content and Antioxidant Activity Evaluation

Table 5 shows the total phenolic content (TPC) and antioxidant activity (DPPH) of the dried and cooled at room temperature pasta samples, obtained by partial replacement of wheat with different levels of hemp seed meal (HSM), which confirms the fact that hemp seed protein presents strong antioxidant effects as the values of both TPC and DPPH significant increase (*p* < 0.05) compared to the control sample. This fact was explainable because hemp seed is rich in natural antioxidants and phenolic compounds [36]. Similar data have also been reported by others for cookies [37] or gluten-free crackers [38] in which different levels of hemp seed meal had been incorporated.

#### 3.4.6. Pasta Sensory Characteristics

The obtained data for sensory characteristics are shown in Figure 5. According to the results, the replacement of wheat grain flour with 20% HSM (HSM_20) led to the lowest scores for all sensory characteristics (global acceptability, aspect, color, smell, taste, texture and aroma) compared to the other samples, concluding that high amounts of hemp seed meal enrichment are not acceptable from the sensory point of view. On the other hand, the most appreciated pasta sample was HSM_15 with 15% HSM substitution. In this case, panelists noticed a well-balanced taste and smell, appealing dark green color and good texture. The control sample was also very appreciated during sensory evaluation, this fact confirming the increased interest of the panelists regarding the consumption of conventional types of pasta, without any other supplementations.

#### 3.4.7. Effect of HSM Enrichment on Pasta Compositional Analysis

The nutritional value of pasta samples formulated by partial replacement of wheat flour with different levels of HSM is shown in Table 6. According to the data obtained, the substitution of wheat with HSM significantly increased (*p* < 0.05) the protein, fat and ash of the pasta samples, whereas the carbohydrate content decreased. This fact is explainable because HSM presents more fat, protein and ash compared to the wheat flour sample.

#### 3.4.8. Principal Component Analysis between Characteristics

The associations between characteristics and pasta samples can be seen in Figure 6 by means of Principal Component Analysis (PCA).

The first principal component explains 57.65% of the total variance, whereas the second explains 20.97%. PC1 was closely associated with pasta technological characteristics, color data of dry and cooked pasta, loss tangent, sensory characteristics except for aspect, textural characteristics except for cohesiveness and nutritional pasta characteristics, while PC2 was associated with dough empirical rheological properties during extension and dynamic module ones, cooking loss, moisture, cohesiveness, sensory pasta and aspect. Pasta samples that contain high levels of HSM substitutes of wheat flour are closer related to each other by being located on the right side of the PCA graph, while the control sample and HSM_5 are placed on the left part of the graph alongside the PC2 component on the PCA. The placement of control sample and HSM_5 on the PCA plot indicates that these pasta samples are very distinctive compared to the samples with higher levels of HSM in the pasta recipe.

## 4. Discussion

Compared to the control sample, hemp protein flour significantly changed (*p* < 0.05) the dough’s empirical rheological properties during extension. At low levels of HSM enrichment, dough has more strength due to an increased value of tenacity and baking strength. This may be because HSM presents a high protein amount, which may prevent the complete hydration of the rest of the dough compounds, mainly starch. In addition, besides proteins, HSM contains other components that may change water binding, affecting the dough rheological properties. Similar data have been reported by Korus et al. for bread and cookie dough where hemp seed proteins were used [37,39]. Contrary to this, when hemp flour was used in dough recipes, the same authors reported a weakening effect on the dough rheological properties. Except baking strength, all the Alveograph dough rheological characteristics did not present significant differences (*p* < 0.05) for the samples with 5% and 10% HSM incorporated into the dough recipe. These data are in agreement with those reported by others who have also concluded that dough rheological characteristics are not significantly different between samples with these levels of hemp meal enrichment [40,41]. When high levels of HSM were added to the dough recipe by replacing the equivalent amount of wheat flour, the baking strength and dough extensibility began to decrease, probably due to the gluten dilution. Moreover, at 20% HSM replacement in wheat flour, the dough tenacity significantly decreased, probably due to the fact that the Alveograph test was made at constant hydration according to the standard method. According to different authors, hemp flour led to a decrease in water absorption value, especially when high levels were added to dough recipe [40,41,42]. Using the same amount of water in the Alveograph method, the dough from the HSM-wheat flour mix may present a slight water excess, which can lead to lower tenacity and baking strength values.

According to the dough’s fundamental rheological data, the control sample presented the highest values of G’ and G” in all frequency ranges. At low levels of HSM substitutions in wheat flour, the decrease in dynamic modules are more evident, whereas at high levels their values begin to increase, the experimental results being similar with those reported by other authors [31]. This behavior may be caused by the water capacity of binding the dough compounds. Hemp protein flours contain proteins and also other compounds such as starch with different properties than those of wheat flour and fibers that modify water absorption in the dough system, etc. [43]. It is possible that when high levels of HSM were added to the wheat flour, a strengthening effect occurred if there was not enough water to hydrate all compounds from the dough system [44]. Loss tangent (tan δ), which is the ratio between viscous and elastic dough components [45], was lower than 1 for all the samples, indicating a viscous–elastic behavior even when high levels of HSM were used to partially replace the wheat flour in the pasta dough [46]. However, when wheat flour was partially replaced with HSM, the tan δ began to decrease, probably due to the gluten dilution from the dough system. Gluten is mainly responsible for the dough viscous–elastic behavior, due to its compounds, such as gliadin and glutenine, which become associated during mixing and form gluten [47]. Wheat grains are the only cereals that can form gluten during mixing [48]. By HSM incorporation, the gluten amount from the dough system was reduced, as was its viscous-elasticity, represented by tan δ values.

As the temperature rose, the G’ and G” values significantly changed. At the beginning, these values decreased due to protein weakening, probably due to the proteolytic activity that began to act on proteins followed by an increase due to the starch gelatinization phase. The G’ and G” values were lower for the control sample and higher for the samples with HSM incorporated into the dough recipe. This may be due to the fiber content of hemp seed proteins, which may increase the viscosity [43]. A similar dough behavior has also been reported by Hrušková et al., who affirmed that HSM enrichment in wheat flour might increase maximum dough viscosity to Amylograph [49]. However, when high levels of HSM were added to the dough recipe, the dynamic modules began to decrease, probably due to the gluten dilution from the dough system. The variation of fundamental rheological characteristics with temperature can also be seen from tan δ, whose values decrease with the increasing level of HSM in wheat flour. This clearly indicates a more viscous behavior of dough samples than elastic ones due to a decreasing amount of gluten from the dough system as a consequence of wheat partial replacement with hemp seed meal (HSM) [46].

The dough microstructure analysis showed that the binding forces between the protein matrix and the starch granules are different according to the HSM incorporation in the pasta samples. Several structural changes can be observed: the appearance of cracks on the outer and inner surfaces, which can be caused due to the contraction or the increase tensions inside the dough during the drying process; the development of alveoli inside the swelling starch granules on the inner surface of pasta samples; agglomerations of starch granules, which can also be observed in the case of pasta samples with 10% and 15% HSM enrichment and not least irregularly distributed starch granules on the whole area surfaces. This change in the pasta structure indicates that the drying process is decisive in pasta manufacturing, as it directly influences pasta cooking behavior. Samples HSM_15 and HSM_20 containing the highest HSM levels presented smoother surfaces compared to HSM_5 and HSM_10, which means better resistance to heat treatment, as a porous network structure is easier for water to penetrate while cooking [22]. The major change in pasta microstructure is reflected in the appearance of cavities of different sizes and shapes, as well as the appearance of starch granules in different stages of gelatinization.

In the case of a new innovative type of pasta, the cooking quality of the product is strictly correlated with pasta technological parameters in terms of optimal cooking time, water absorption, cooking loss and swelling index. On the other hand, technological properties are affected by both the raw materials used and the pasta production process. From the sensory point of view, undercooked pasta with a suitable firm texture, referred to as “al dente” is more appreciated by consumers [50]. In addition, the optimal cooking time (OCT) influences the consistency and firmness of pasta in the course of cooking [20,51,52]. The replacement of wheat with HSM in pasta led to an increase in OCT and CL, while WA and SI decreased compared to the control sample. Similar results were obtained by Ungureanu Iuga [23] for a corn-based pasta and Bustos et al. [53] for a berry-enriched pasta, as the pasta enrichment led to gluten weakening. The weakening of the gluten network is the result of HSM replacement in wheat flour, which has high protein content but does not contain gluten. Regarding the cooking loss, values did not exceed 8%, similar results being reported by Teterycz et al. [6], where the cooking loss increased with the addition of hemp raw materials, especially with hemp cake. A similar tendency has also been reported in other studies when millet flour, pearl millet flour and carrot pomace powder were used [54]. As the replacement of wheat flour increases, the binding power of gluten decreases. The swelling Index (SI) of the pasta samples decreased with the increased levels of HSM enrichment due to a weaker gluten structure and, respectively, a smaller amount of gelatinized starch.

Compared to the cooked pasta, the dry pasta samples showed lower brightness (L*) and higher redness (a* values). Such parameters were obtained due to the Maillard reactions, as the process of drying requires relatively high temperatures and the process itself takes several hours. For pasta with 20% HSM, the drying and cooking heat treatments led to significant changes of the pasta color. During sensory evaluation, panelists noticed an unpleasant color, which influenced the general acceptability of the product.

The same as in the case of pasta enriched with hemp raw materials, as described by Teterycz et al., 2021 [6], the partial replacement of wheat grain flour with HSM did not influence the springiness of the final product. All samples were very springy. Pasta samples became firmer with the increase in HSM enrichment. Adding hemp seed meal reduced the quantity of wheat flour, and therefore the gluten network became weaker. As the gluten network got weaker, the textural parameters showed a significant increase in adhesiveness and stickiness. Pasta with 20% HSM (HSM_20) was the stickiest. The stickiness is related to the amount of amylose leached from the gelatinized starch granules [9]. This characteristic was also noticed by the consumer panel by chewing the sample. Regarding the cohesiveness, there were no significant changes observed.

Sensory profile evaluation confirmed the interest of consumers regarding the innovative pasta formulations, as panelists noticed the appealing green color and pleasant taste of HSM_15 pasta sample with wheat flour substituted with hemp seed meal in proportions of 15%, which presented the optimal sensory parameters. On the other hand, lower levels of HSM negatively influenced the panelists’ perception of the product, as they noticed the color loss, an unpleasant stickiness of the samples while chewing and pasta samples were also tasteless compared to the higher levels of HSM enrichment.

From a nutritional point of view, HSM-enriched pasta significantly increased (*p* < 0.05) its total protein, fat content, phenolic content and antioxidant activity. The nutritional improvements of cereal products in which hemp seed meal have been incorporated have also been reported by others [40,41,55]. According to them, the hemp seed meal enrichment significantly helped to increase the total protein and fat content even more with the increasing level of hemp seed meal partial substitution in wheat flour. In addition, an increase in antioxidant activity and total phenolic content have been reported by others [37,55] due to the high amount of polyphenols from hemp seed meal that included ferulic acid, epicatechin and protocatechuic acid.

The PCA plot shows a clear dissimilarity between the pasta samples. The control sample positioned at the bottom left of the graph is closely related to swelling index, water absorption, fracturability, adhesiveness, pasta luminosity and sensory characteristics such as smell, taste and texture. This sample is closely related to the pasta sample with the lowest level of HSM replacement in wheat flour, whereas the samples with high levels of HSM incorporated into the pasta recipe are more closely associated with being more related to the nutritional characteristics of pasta samples. Significant correlations (*p* < 0.05) were obtained between dough extensibility and index of swelling (r = 0.99) due to the fact that both are expressions of dough rheological properties during extension [56], between baking strength and cooking loss (r = 0.937) and between elastic modulus and cohesiveness (r = 0.982). Similar results have been pointed out by Romero et al., who concluded that pasta dough with a more elastic and stronger structure network has the highest cohesiveness [57]. Loss tangent is strongly correlated (*p* < 0.05) with textural pasta characteristics firmness (r = −0.944), stickiness (r = 0.926) and pasta fracturability (r = 0.890). This association may be due to the gluten dilution in the dough system from using the HSM, a non-gluten ingredient that partial substitutes wheat flour in the pasta recipe. Total phenolic content and antioxidant activity are closely significantly (*p* < 0.05) associated with each other (r = 0.955), being positively (*p* < 0.05) correlated with fat (r = 0.933; r = 0.927) ash (r = 0.922; 0.903) and negatively (*p* < 0.05) correlated with dry pasta luminosity (r = −0.952; r = −0.953), pasta’s luminosity (r = −0.977; r = −0.947) and carbohydrates content (r = −0.891; r = −0.918). The HSM replacement in wheat flour increased the pasta nutritional characteristics due to its much higher protein, ash, fat content, antioxidant activity and total phenolic content compared to wheat flour. Therefore, these characteristics are significantly correlated. In addition, HSM presents natural pigments that obviously change the pasta color properties [43]. Increases of the darkness of cereal food product color by hemp meal addition to the food products have also been reported by others [40,42,55].

## 5. Conclusions

Hemp protein pasta is an innovative food product that successfully diversifies the varieties of plant-based protein types of pasta that are currently exposed on the global market. Pasta enriched with hemp seed meal can be classified as a functional product due to the benefits this product provides for the human diet, mostly related to the protein content, antioxidant potential and great nutritional value. Regarding the experimental results, the replacement of wheat grain flour with different levels of hemp seed meal led to an increased firmness but higher stickiness and adhesiveness, lower fracturability and better dough extensibility. In addition, a decrease in cooking quality was observed in terms of higher cooking loss compared to the control sample. SEM micrograms highlighted that, during drying, pasta structure suffers gluten weakening that leads to numerous cavities, fissures and the agglomeration of proteins around the starch particles. In addition to its valuable nutritional value, an increased phenolic content (TPC) and antioxidant activity of enriched protein pasta confirm the functional potential of the product, as the consumption of hemp seed-supplemented products is associated with reducing the risks of multiple stress oxidative diseases. From a sensory point of view, HSM-enriched pasta was well appreciated, as panelists noticed an appealing dark green color and pleasant taste of the product (HSM_15).

The experimental data obtained in this study will definitely be useful for other researchers, as the hemp meal use in different food products is currently not studied enough but represents a high interest topic worldwide. The experimental results will also be used in our further investigations.

## Figures and Tables

**Figure 1 foods-12-01774-f001:**
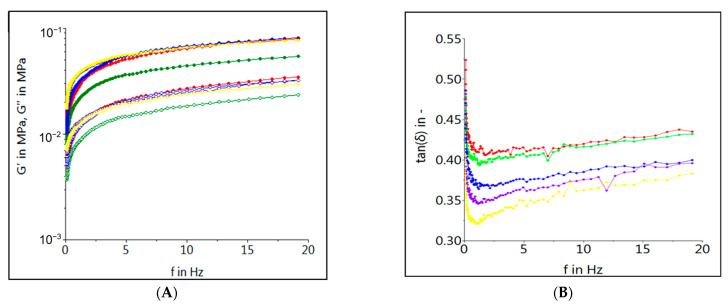
Frequency dependent evolution of storage modulus G’ (showed by solid symbols), loss modulus G” (showed by empty symbols) (**A**) and tan δ (**B**) of pasta dough samples with different levels of HSM ((-•- Control; -•- 5%; -•- 10%; -•- 15%; -•- 20%) incorporated in wheat flour.

**Figure 2 foods-12-01774-f002:**
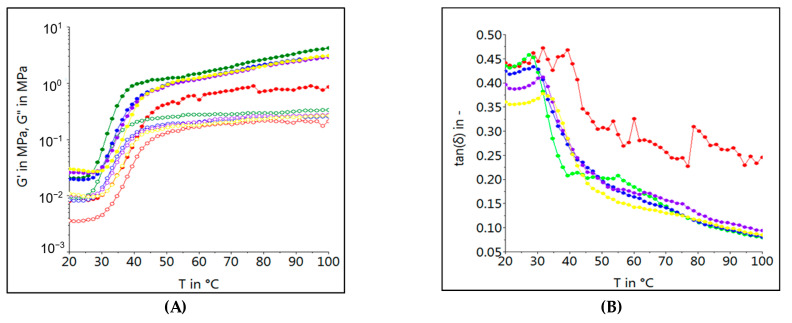
Temperature dependent evolution of storage modulus G’ (showed by solid symbols), loss modulus G” (showed by empty symbols) (**A**) and tan δ (**B**) of pasta dough samples with different levels of HSM (-•- Control; -•- 5%; -•-10%; -•- 15%; -•- 20%) substitution in wheat flour.

**Figure 3 foods-12-01774-f003:**
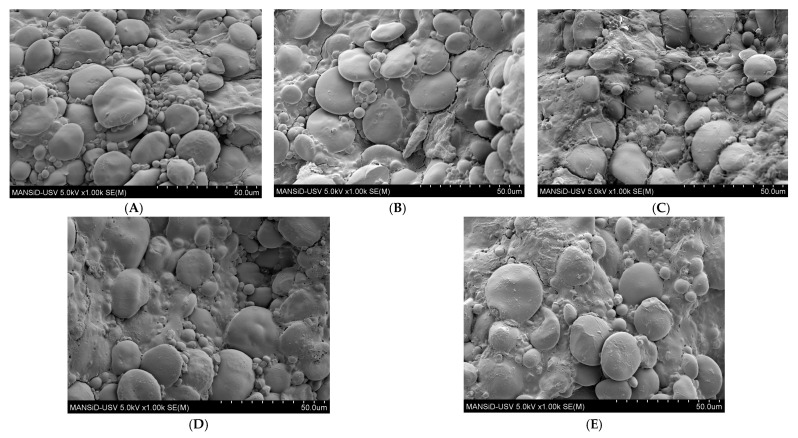
Scanning electron micrographs (SEM) of dried pasta with different levels of HSM substitution in wheat flour: 0% (**A**), 5%(**B**), 10%(**C**), 15%(**D**), 20%(**E**).

**Figure 4 foods-12-01774-f004:**
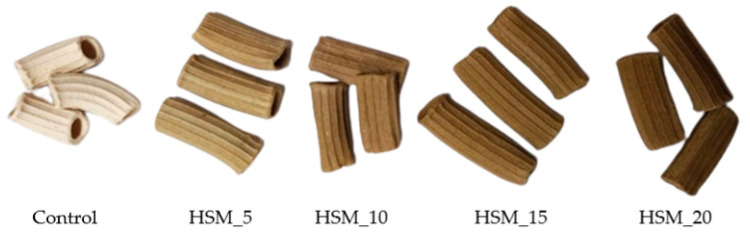
Dry pasta samples with different levels of HSM enrichment: 0% (Control), 5%(HSM_5), 10%(HSM_10), 15%(HSM_15), 20%(HSM_20).

**Figure 5 foods-12-01774-f005:**
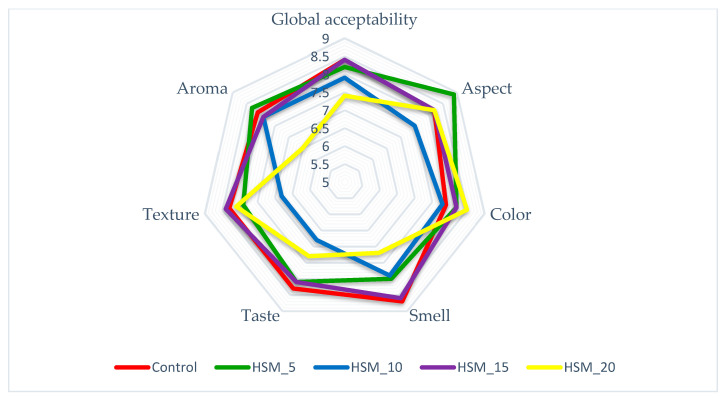
Sensory characteristics of pasta enriched with hemp seed meal.

**Figure 6 foods-12-01774-f006:**
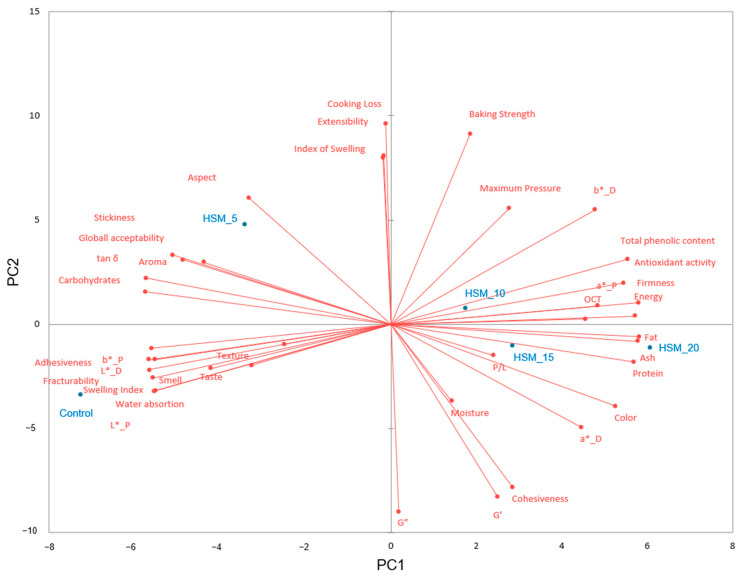
Principal Component Analysis (PCA).

**Table 1 foods-12-01774-t001:** Alveograph properties of pasta dough samples enriched with HSM.

Sample	P (mm)	L (mm)	G (mm)	W (10^−4^ J)	P/L (adim)
Control	43 ± 1.52 ^a^	31 ± 0.57 ^a^	12.4 ± 0.45 ^a^	50 ± 0.57 ^a^	1.39 ± 0.07 ^a^
HSM_5	88 ± 1.52 ^c^	55 ± 1.52 ^c^	16.5 ± 0.17 ^c^	157 ± 2.00 ^e^	1.6 ± 0.06 ^a^
HSM_10	86 ± 2.51 ^c^	57 ± 2.00 ^c^	16.8 ± 0.35 ^c^	139 ± 2.08 ^d^	1.51 ± 0.10 ^a^
HSM_15	105 ± 4.04 ^d^	28 ± 2.00 ^a^	11.7 ± 0.05 ^a^	121 ± 1.00 ^c^	3.75 ± 0.41 ^b^
HSM_20	66 ± 2.51 ^b^	37 ± 13.00 ^b^	13.5 ± 0.35 ^b^	97 ± 1.52 ^b^	1.78 ± 0.16 ^a^

P—maximum pressure; L—dough extensibility; G—index of swelling; W—baking strength; P/L—configuration ratio of the Alveograph curve. The results are the mean ± standard deviation (n = 3). Dough samples containing hemp seed meal—HSM; ^a–e^—mean values in the same column followed by different letters are significantly different (*p* < 0.05).

**Table 2 foods-12-01774-t002:** Technological data of pasta with different levels of HSM in wheat flour.

Pasta Sample	OCT (min)	WA (%)	CL (%)	SI
Control	9.00 ± 0.10 ^a^	113.66 ± 1.15 ^d^	3.28 ± 0.02 ^a^	1.39± 0.00 ^d^
HSM_5	9.26 ± 0.05 ^b^	101.33 ± 1.52 ^c^	3.43 ± 0.01 ^c^	1.27 ± 0.01 ^c^
HSM_10	9.23 ± 0.05 ^b^	98.00 ± 1.00 ^b^	3.39 ± 0.01 ^c^	1.25 ± 0.11 ^bc^
HSM_15	9.23 ± 0.11 ^b^	96.00 ± 1.00 ^ab^	3.34 ± 0.01 ^b^	1.23 ± 0.01 ^b^
HSM_20	9.96 ± 0.06 ^c^	93.33 ± 0.57 ^a^	3.30 ± 0.15 ^ab^	1.16 ± 0.02 ^a^

OCT—optimal cooking time, WA—water absorption, CL—cooking loss, SI—swelling index. The results are the mean ± standard deviation (n = 3). Dough samples containing hemp seed protein—HSM; ^a–d^—mean values in the same column followed by different letters are significantly different (*p* < 0.05).

**Table 3 foods-12-01774-t003:** Color data of dry and cooked pasta with different levels of HSM in wheat flour.

Pasta Samples	Dry Pasta			Cooked Pasta		
	L*	a*	b*	L*	a*	b*
Control	71.51 ± 2.28 ^c^	1.12 ± 0.08 ^ab^	16.62 ± 0.32 ^a^	68.32 ± 3.12 ^c^	−0.08 ± 0.05 ^a^	16.66 ± 0.68 ^c^
HSM_5	57.77 ± 2.95 ^b^	0.78 ± 0.10 ^a^	20.42 ± 0.39 ^b^	51.63 ± 0.92 ^b^	0.99 ± 0.14 ^b^	14.60 ± 0.17 ^b^
HSM_10	45.95 ± 0.05 ^a^	1.54 ± 0.03 ^bc^	21.58 ± 0.25 ^b^	43.84 ± 0.83 ^a^	0.69 ± 0.10 ^b^	11.95 ± 0.40 ^a^
HSM_15	43.74 ± 0.64 ^a^	1.28 ± 0.36 ^bc^	20.47 ± 0.83 ^b^	43.55 ± 0.23 ^a^	2.28 ± 0.13 ^c^	11.82 ± 0.42 ^a^
HSM_20	43.01 ± 1.01 ^a^	1.66 ± 0.09 ^c^	21.13 ± 0.70 ^b^	41.06 ± 1.83 ^a^	1.95 ± 0.20 ^c^	11.81 ± 0.36 ^a^

L*—luminosity, a*—red-green intensity, b*—yellow-blue intensity. The results are the mean ± standard deviation (*n* = 10). Pasta samples containing hemp seed meal—HSM: ^a–c^, mean values in the same column followed by different letters are significantly different (*p* < 0.05).

**Table 4 foods-12-01774-t004:** Pasta textural evaluation.

			Pasta Dough			Dry Pasta
Sample	Firmness (N)	Adhesiveness (J)	Springiness (%)	Cohesiveness (adim)	Stickiness (%)	Fracturability (N)
Control	13.17 ± 5.07 ^a^	−15.97 ± 0.02 ^e^	0.99 ± 0.02 ^a^	0.36 ± 0.02 ^b^	−41.98 ± 0.56 ^b^	53.08 ± 0.38 ^b^
HSM_5	13.89 ± 0.52 ^b^	−53.58 ± 0.01 ^d^	0.99 ± 0.01 ^a^	0.32 ± 0.01 ^a^	−37.77 ± 0.04 ^c^	44.92 ± 0.33 ^b^
HSM_10	14.69 ± 15.07 ^c^	−63.41 ± 0.02 ^c^	0.99 ± 0.01 ^a^	0.37 ± 0.02 ^b^	−60.49 ± 0.01 ^a^	38.29 ± 0.34 ^b^
HSM_15	14.75 ± 7.71 ^c^	−65.03 ± 0.04 ^b^	0.99 ± 0.02 ^a^	0.36 ± 0.02 ^b^	−60.45 ± 0.02 ^a^	41.03 ± 0.33 ^a^
HSM_20	15.01 ± 0.50 ^d^	−104.59 ± 0.06 ^a^	1.02 ± 0.01 ^b^	0.37 ± 0.02 ^b^	−59.81 ± 0.02 ^a^	35.26 ± 0.19 ^a^

Results are the mean ± standard deviation (*n* = 3). Dough samples containing hemp seed meal—HSM: ^a–e^ mean values in the same column followed by different letters are significantly different (*p* < 0.05).

**Table 5 foods-12-01774-t005:** Total phenolic content and antioxidant activity of pasta samples.

Pasta Sample	TPC (μg GAE/g)	DPPH (%)
Control	13.34 ± 0.17 ^a^	18.48 ± 0.53 ^a^
HSM_5	16.12 ± 0.40 ^b^	21.57 ± 0.39 ^b^
HSM_10	16.68 ± 0.04 ^bc^	22.06 ± 0.10 ^b^
HSM_15	17.02 ± 0.14 ^c^	24.16 ± 1.00 ^b^
HSM_20	18.14 ± 0.04 ^d^	24.00 ± 0.16 ^c^

TPC—total phenolic content, DPPH—antioxidant activity; ^a–d^—mean values in the same column followed by different letters are significantly different (*p* < 0.05).

**Table 6 foods-12-01774-t006:** Proxymate analysis of pasta samples enriched with hemp seed meal (HSM).

Pasta Samples	Moisture (%)	Protein (%)	Fat (%)	Ash (%)	Carbohydrates (%)	Energy (kcal/100 g)
Control	11.22 ± 0.02 ^b^	9.52 ± 0.01 ^a^	0.83 ± 0.01 ^a^	0.47 ± 0.01 ^a^	77.98 ± 0.04 ^e^	357 ± 0.14 ^a^
HSM_5	11.18 ± 0.01 ^a^	10.24 ± 0.02 ^b^	1.21 ± 0.01 ^b^	0.59 ± 0.01 ^b^	76.82 ± 0.02 ^d^	359 ± 0.01 ^ab^
HSM_10	11.27 ± 0.01 ^d^	12.13 ± 0.01 ^c^	1.67 ± 0.02 ^c^	0.76 ± 0.01 ^c^	74.17 ± 0.01 ^c^	360 ± 0.01 ^b^
HSM_15	11.25 ± 0.01 ^c^	13.84 ± 0.01 ^d^	1.95 ± 0.03 ^d^	0.84 ± 0.02 ^d^	72.12 ± 0.01 ^b^	361 ± 0.01 ^b^
HSM_20	11.20 ± 0.01 ^b^	14.72 ± 0.01 ^e^	2.38 ± 0.04 ^e^	1.03 ± 0.03 ^e^	70.67 ± 0.01 ^a^	363 ± 0.01 ^c^

The results are mean ± standard deviation (*n* = 3). ^a–e^—mean values in the same column followed by different letters are significantly different (*p* < 0.05).

## Data Availability

The datasets generated for this study are available on request from the corresponding author.

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
