# Peer review of "Development and Quality Evaluation of Rigatoni Pasta Enriched with Hemp Seed Meal"

_foods, 2023, doi:10.3390/foods12091774_

Round 1
Reviewer 1 Report
-This manuscript examined the properties of pasta added with hemp seed meals at 5-20 g/100g dry basis. In general, this manuscript lacks novelty despite its extensive results as various techniques were used for analysis.
-The addition of high protein ingredients such as hemp seed meals will of course change the properties of the pasta. Results can be expected. Though, this manuscript provides results that could be used for further research or industrial applications.
-The English needs to be polished by professional editing services. The manuscript length is unnecessarily long, and it can be more concise. The structure of the manuscript should also be edited. Some similar properties e.g. physicochemical properties, physical properties, antioxidant properties etc. should be grouped together, the manuscript can be easy to read and follow.
-Pasta is generic name, please be specific e.g. rigatoni pasta as per the mold used?
-Hemp seed meal should be used instead of hemp seed protein as the authors did not isolate proteins from hemp seed meal. It only contains about 50 g/100 g protein.
-SEM images are unclear due to low magnification being applied (only 260 x).
-SI units should be used throughout the manuscript.
-Check the data and statistical superscripts e.g. Table 4 (Springiness). Is it the same value for all samples?
Author Response
14 April 2023
Dear Referee,
We sincerely appreciate all your valuable comments and suggestions, which helped us in improving the quality of our paper entitled “Development and quality evaluation of pasta enriched with hemp seed protein”. We are extremely grateful for the precious time and effort that you have dedicated to reviewing our manuscript. We modified the manuscript according to referee suggestions. To ensure easy reading of the revised manuscript, we highlighted the changes, using the strikethrough formatting option for all kinds of mistakes and the red colour for added/corrected text.
Please find below, a point-by-point response to all your comments and concerns.
Referee comments: This manuscript examined the properties of pasta added with hemp seed meals at 5-20 g/100g dry basis. In general, this manuscript lacks novelty despite its extensive results as various techniques were used for analysis.
Response: We would like to thank to the referee for his/her close reading of our manuscript. Since there is an ever-growing trend of sustainable products using hemp worldwide and there are no manufacturers in Romania who produce high protein pasta with hemp seed meal, by writing this manuscript, we would also like to support Romanian manufacturers (and not only, this study being addressed and to other pasta producers which wants to enrich their pasta products) by formulating a recipe that presents industrial feasibility and can be reproduced at industrial scale (we used in our study a hemp product produced in Romania, too). The hemp seed market is growing nowadays as hemp seeds are a rich source of protein (including lysine which is in low amount in wheat flour), minerals and fiber, essential fatty acids (including omega 3 which now is deficient in humans diet) in high amount being considered one of the most suitable sources of plant-based protein that also has a relatively low cost and can be easily incorporated into different food products, such as pasta. The multitude of techniques used in this manuscript confirms our great interest in developing hemp-rich products to prove the importance of including hemp in our diet. From the technological point of view our data are complex ones (we even can affirm that are complete ones) since includes rheological, microstructure, uncooked pasta quality characteristics and cooked pasta quality characteristics. However, we find studies on using hemp raw materials in bread and cookies and very few on pasta. Even so, the studies find by us are different than our study which we believe that is more focused on technological behavior of hemp addition on wheat flour having a more industrial approach. More, nowadays is an increasing trend for natural food products (in especial vegan ones) with high nutritional value. Also, a valorization of by-products from the food industry is desired (as our hemp seed meal). We consider that our study (which will be part of a doctoral thesis) is a unique one so far through it complex approach focusing more on hemp use in pasta to an industrial level.
Referee comments: The addition of high protein ingredients such as hemp seed meals will of course change the properties of the pasta. Results can be expected. Though, this manuscript provides results that could be used for further research or industrial applications.
Response: We want to thank once again to the referee for the close reading of our manuscript. Indeed, the addition of high protein ingredients such as hemp seed meals we will change the properties of the pasta. It is well known that any ingredient that is added in wheat flour will change the dough rheological behaviour and final product quality. It is important to know for industrial purpose is these changes are significant or not and of course to what modificiantions we will expect. As we mentioned in conclusion section, the experimental results will be used by us for further investigations, as we plan to formulate other hemp-rich products, by using hemp meal, oil and fibers (what we know so far will be very useful). Of course, other authors can start from our research for further investigations. For such purpose, we find this manuscript extremely helpful. On the other hand, to industrialize this product, it was particularly important to understand how the partial replacement of wheat flour influences the technological behavior through rheological and quality pasta properties (completed also with microstructural analysis) of the pasta samples, as we previously mentioned.
Referee comments: The English needs to be polished by professional editing services. The manuscript length is unnecessarily long, and it can be more concise. The structure of the manuscript should also be edited. Some similar properties e.g. physicochemical properties, physical properties, antioxidant properties etc. should be grouped together, the manuscript can be easy to read and follow.
Response: We want to thank to the referee for his/her suggestions. The whole manuscript was corrected with the help of our colleague, who is a native English-speaker. As you suggested, we changed the structure, according to the Foods journal’s template and grouped together the fracturability data of the dry pasta and textural properties of the pasta dough as both refer to the pasta textural properties. However, we didn’t group together other experimental results, such as technological properties or antioxidant properties of pasta samples because we consider that all of them present distinct information (hope to be oK this structure). We really hope that now the structure and English language of our manuscript is more sustainable for publication in Foods journal.
Referee comments: Pasta is generic name, please be specific e.g. rigatoni pasta as per the mold used?
Response: Thank you for pointing this out. We agree with your comment. We have modified the name, as you suggested, in “rigatoni pasta” and “rigatoni-shaped pasta”, as we used a rigatoni mold for the pasta dough.
Referee comments: Hemp seed meal should be used instead of hemp seed protein as the authors did not isolate proteins from hemp seed meal. It only contains about 50 g/100 g protein.
Response: Thank you for this excellent observation. We found your comment extremely helpful and have revised accordingly.
Referee comments: SEM images are unclear due to low magnification being applied (only 260 x).
Response: Thank you for pointing this out for us. We made once again new SEM analysis. We strongly agree with your comment and uploaded new SEM images with a much higher magnification (x1000) and we truly believe all of them are spectacular. Thank you for your excellent feedback regarding this aspect! We found it very helpful!
Referee comments: SI units should be used throughout the manuscript.
Response: Thank you for highlighting this aspect. We agree with your suggestion and have corrected non SI units throughout the manuscript.
Referee comments: Check the data and statistical superscripts e.g. Table 4 (Springiness). Is it the same value for all samples?
Response: Thank you for pointing this out. We have checked the statistical data and found out that Springiness data differ, but only starting with the third decimal. We have the data in the table only with 2 decimals since all our data are with 2 decimals in our manuscript.
Sincerely,
Codină et al.

Reviewer 2 Report
Please see comment in the attachment.

Author Response
14 April 2023
Dear Referee,
We sincerely appreciate all your valuable comments and suggestions, which helped us in improving the quality of our paper entitled “Development and quality evaluation of pasta enriched with hemp seed protein”. We are extremely grateful for the precious time and effort that you have dedicated to reviewing our manuscript. We modified the manuscript according to referee suggestions. To ensure easy reading of the revised manuscript, we highlighted the changes, using the strikethrough formatting option for all kinds of mistakes and the red colour for added/corrected text.
Please find below, a point-by-point response to all your comments and concerns.
Referee comments: From methodology and results, the material used in this study should not be called hemp seed protein, because it still contained large amount of fat and carbohydrate. It did not from the protein extraction process.
Response: Thank you for your excellent observation. We agree with this comment. We changed the term and the abbreviation respectively to: hemp seed meal (abbreviated HSM) according to yours suggestions.
Referee comments: What is the particle size of the powder? Please include this information into manuscript.
Response: Thank you for pointing this out. In this manuscript we have used hemp protein powder with fine particles. The manufacturing company who sells it (Cannah, Romania) affirms that the particle size of their powder is medium around 200-236 µm. We completed now in the manuscript this information.
Referee comments: From dough characteristic analysis, the study use the same hydration for all recipes. This should be reconsidered. To obtain the good dough quality, water addition should be optimized in accordance to amount of added protein, starch, and fiber. In this study, 20% addition of hemp seed might not be that bad, if water addition was optimized.
Response: We really wanted to thank to the referee for the close reading of our manuscript. Except Alveograph data all the analysis were made to different hydration capacity. We mentioned now in the manuscript to the 2.2. section the optimum dough hydration levels for each mixes. However, to Alveograph we used the standard method, namely AACC 54-30A, ICC 121, or ISO 5530/4. According to the standard method, the water addition is due to the mix humidity. In our country the Alveograph is the most used rheological method. Even if the hydration capacity of wheat flour vary the amount of water added in the mixer is according to the flour humidity. We explain in the manuscript that this strange behaviour of the dough to the extension tests are due to the fact that we used the same amount of water in the mixer becouse that we used the standard procedure. For industrial propose, these informations are useful since all the operators are working according to the standard methods.
Referee comments: Formulation of pasta dough should be explained in the methodology.
Response: Agree. We have added the suitable explications in the methodology (section 2.2.) as you suggested.
Referee comments: Methodology to prepare dough, fresh pasta and dried pasta should be clearly stated.
Response: We strongly agree with this comment. Therefore, we have stated in section 2.2. the pasta dough and dry pasta formulation processes, including all necessary details to ensure data reproducibility.
Referee comments: Did SEM determine the dried pasta structure? If so, their results should be included in dough properties. In fact, SEM should be included in the dried pasta quality.
Response: Thank you for your suggestion. Yes, SEM analysis was performed on dry pasta microstructure. We agree that SEM analysis refers to the dry pasta quality, however in case of our manuscript it plays an important role in appreciating textural, rheological and technological properties of pasta samples (by presenting the visual effects of pasta mechanical and heat processing).
Referee comments: Why was DPPH selected to determine antioxidant activity of the pasta? Why not others (such as ABTS...)? Please insert some explanation in the methodology. In addition, EC50 should be also estimated.
Response: The pasta antioxidant activity was performed using DPPH, as this method was efficiently applied by us and our colleagues in other cases of enriched pasta formulations, using such materials as grape peels, spent grain, etc. In our faculty is the only method used for the antioxidant activity evaluation.
Referee comments: Were the results of antioxidant analysis from the dried pasta or cooked pasta? Loss of phenolic compounds and antioxidant activity was always found during cooking.
Response: Thank you for your question. Your feedback helped us a lot in improving the paper. We have specified in our revised manuscript that the determination of antioxidant properties of pasta samples were done using dried, and cooled at the room temeprature, pasta. We sincerely appreciate your observation.
Referee comments: Why was yellowness (b* value) of pasta with HSP clearly reduced after cooking? , but not happened with the normal pasta.
Response: These changes may be due probably due to the partial pigments released in the boiling water after cooking as we mentioned in our manuscript. After boiling, pasta samples presented a slight discoloration, which modify the a* and b* parameters. These explanations are presented in the manuscript.
Referee comments: To claim 'high protein food', how much should be the protein content in the food? Please include this information into the manuscript.
Response: We really wants to thank to the referee for the close reading of our manuscript, When it comes to high protein pasta market, all manufacturers that produce pasta with at least 10 g of protein per serving label the product as “high protein pasta”. According to this fact and also considering the ingredients commonly used for high protein pasta, we have concluded that the content should be more than 10g of protein per serving (the serving size is 56 g). However, now, when referee point us our 'high protein food' claim we read the legislations and we found out that 'a food is high in protein, and any claim likely to have the same meaning for the consumer, may only be made where at least 20% of the energy value of the food is provided by protein'. Our pasta provide only 16.6% of the total eneregy value from proteins. Therefore we concluded that our pasta are only enriched in proteins and is not a high protein food. We deleted the claim of a high protein food and we changed with enriched in proteins. We really thanks to the referee for helping us not to have misatkes in our manuscript.
Referee comments: Materials and Methodology should be stated before showing the results
Response: Thank you for pointing this out. We found your comment very helpful. We have changed the structure of the revised manuscript according to the Foods template and as you suggested.
Sincerely,
Codină et al.

Reviewer 3 Report
The qualitative features of wheat pasta enriched with hemp seed protein were characterized in this work. Standard analyzes were performed on both the pasta dough and the finished product. The work is written concisely. Here are some specific comments.
Please be precise in naming the samples - the wheat flour was substituted with hemp seed protein in the amount of 5-20% but not the HSP was added (the Authors use term 5-20% additions); alternatively, please specify the sample naming system.
Please remove the last two sentences of Introduction, since they refer to the experimental results.
Line 165: There should be Figure 3 instead of Figure 1.
Table 4 and Table 5. Please convert the non-SI units into SI ones.
Line 240: There should be Table 5 instead of Table 4.
Lines 356-357: How would you explain the protein weakening?
Lines 367-368: The phrase "the gluten decrease amount form the dough system" is not understable.
Section 4. Please add explanation of pasta dough samples preparation before Section 4.1.
Line 555: The formula is not complete.
Title of the Section 4.7. It should refer to both pasta dough and pasta textural parameters.
Section 4.7. Please describe how was the pasta dough sample formed for the compression test. Please define dimension of the dry pasta sample for the fracturability test.
Author Response
14 April 2023
Dear Referee,
We sincerely appreciate all your valuable comments and suggestions, which helped us in improving the quality of our paper entitled “Development and quality evaluation of pasta enriched with hemp seed protein”. We are extremely grateful for the precious time and effort that you have dedicated to reviewing our manuscript. We modified the manuscript according to referee suggestions. To ensure easy reading of the revised manuscript, we highlighted the changes, using the strikethrough formatting option for all kinds of mistakes and the red colour for added/corrected text.
Please find below, a point-by-point response to all your comments and concerns.
Referee comments: The qualitative features of wheat pasta enriched with hemp seed protein were characterized in this work. Standard analyzes were performed on both the pasta dough and the finished product. The work is written concisely. Here are some specific comments.
Response: Thank you very much for your appreciation. Your valuable feedback helped us improving this manuscript.
Referee comments: Please be precise in naming the samples - the wheat flour was substituted with hemp seed protein in the amount of 5-20% but not the HSP was added (the Authors use term 5-20% additions); alternatively, please specify the sample naming system.
Response: Thank you for your excellent observation. We indeed made this mistake several times (used the term additions instead of replacement). We agree with your comment and have revised the manuscript accordingly. Also we specified and changed the sample naming system, as other reviewers suggested into HSM (abbreviated Hemp Seed Meal) followed by the substituted amount of wheat (HSM_5, HSM_10,HSM_15,HSM_20 and the control sample) with HSM (5-20%), as we did not isolate the protein from hemp.
Referee comments: Please remove the last two sentences of Introduction, since they refer to the experimental results.
Response: Thank you for your observation. As you suggested, we have deleted the last 2 sentences of the introduction.
Referee comments: Line 165: There should be Figure 3 instead of Figure 1.
Response: Thank you for noticing this mistake, we have corrected it.
Referee comments: Table 4 and Table 5. Please convert the non-SI units into SI ones.
Response: Thank you for pointing this out. We have converted the non-SI units in table 4 and table 5 (for the fracturability), but as other reviewer suggested we grouped up this 2 tables together, as they both refer to the textural parameters of the pasta samples.
Referee comments:: Line 240: There should be Table 5 instead of Table 4.
Response: Thank you for your excellent observation. We corrected. Also, we grouped together table 4 and table 5, so the manuscript can be easy to read and follow.
Referee comments::: Lines 356-357: How would you explain the protein weakening?
Response: The protein weakening is due to the temperature rise. As the temperature rise, dough temperature begins to increase and the proteases from the dough system to act which will weak the proteins.
Referee comments::: Lines 367-368: The phrase "the gluten decrease amount form the dough system" is not understable.
Response: Analyzing the context where the phrase was used, we were explaining that the amount of gluten decreased as a consequence of wheat partial replacement with HSM, which influenced the dough system. We also rephrased the sentence.
Referee comments:: Section 4. Please add explanation of pasta dough samples preparation before Section 4.1.
Response: Thank you for helping us improving the structure of our manuscript. We have explained the formulation process and specified the amounts used of each ingredient for pasta dough preparation. As we had the pasta formulation process explained at section 4.4 we switched the place, as you suggested, and insert it before section 4.1. As Foods journal template has Materials and Methods section before Results, the actual number of pasta preparation section is 2.2. (right after 2.1. Materials).
Referee comments:: Line 555: The formula is not complete.
Response: Thank you for noticing. We have completed the formula.
Referee comments: Title of the Section 4.7. It should refer to both pasta dough and pasta textural parameters.
Response: We agree with this comment. We have changed it.
Referee comments: Section 4.7. 1Please define dimension of the dry pasta sample for the fracturability test.
Response: Thank you for pointing this out. We have completed the manuscript with the specific pasta size. For the fracturability test we used dry pasta samples 45 mm long with a 16 mm diameter.
Sincerely,
Codină et al.

Round 2
Reviewer 2 Report
No more comment.